# Observation of plastic ice VII by quasi-elastic neutron scattering

Maria Rescigno[1,2], Alberto Toffano[3,4], Umbertoluca Ranieri[1,5,10], Leon Andriambariarijaona[1], Richard Gaal[2], Stefan Klotz[6], Michael Marek Koza[7], Jacques Ollivier[7], Fausto Martelli[4,8], John Russo[1], Francesco Sciortino[1], Jose Teixeira[9] & Livia Eleonora Bove[1,2,6 ✉]

Water is the third most abundant molecule in the universe and a key component in the interiors of icy moons, giant planets and Uranus- and Neptune-like exoplanets[1–3]. Owing to its distinct molecular structure and flexible hydrogen bonds that readily adapt to a wide range of pressures and temperatures, water forms numerous crystalline and amorphous phases[4–6]. Most relevant for the high pressures and temperatures of planetary interiors is ice VII (ref. 4), and simulations have identified along its melting curve the existence of a so-called plastic phase[7–12] in which individual molecules occupy fixed positions as in a solid yet are able to rotate as in a liquid. Such plastic ice has not yet been directly observed in experiments. Here we present quasi-elastic neutron scattering measurements, conducted at temperatures between 450 and 600 K and pressures up to 6 GPa, that reveal the existence of a body-centred cubic structure, as found in ice VII, with water molecules showing picosecond rotational dynamics typical for liquid water. Comparison with molecular dynamics simulations indicates that this plastic ice VII does not conform to a free rotor phase but rather shows rapid orientational jumps, as observed in jump-rotor plastic crystals[13,14]. We anticipate that our observation of plastic ice VII will affect our understanding of the geodynamics of icy planets and the differentiation processes of large icy moons.

Water's intricate phase diagram contains more than 20 crystalline[5,15] and 3 amorphous[6] phases, with further ice phases predicted to occur under extreme pressure and temperature ($P–T$) conditions and awaiting experimental confirmation[16–18]. At pressures exceeding 2 GPa, the phase diagram simplifies to three primary water phases: proton-disordered ice VII (ref. 19), its proton-ordered counterpart ice VIII (ref. 20), and the atomic phase ice X (ref. 21), which share a common cubic oxygen lattice—with a slight tetragonal distortion in ice VIII—and differ in how hydrogen atoms are arranged. In ice VII, the oxygen atoms form a body-centred cubic structure where each oxygen atom connects to four hydrogen atoms through two covalent and two hydrogen bonds; the resultant proton-disordered network comprises two independent interpenetrating subnetworks, each topologically equivalent to cubic ice I. As temperature decreases, this network transitions to proton-ordered ice VIII, marked by a slight tetragonal distortion. At higher pressures, the hydrogen bonds symmetrize to give ice X, the distinctive atomic form of ice.

As both pressure and temperature increase and hydrogen becomes more mobile, new exotic ice phases emerge. This includes hybrid phases that show both solid- and liquid-phase properties, such as the predicted[22] and indirectly observed[23,24] superionic state with oxygen atoms occupying the ice X lattice positions while protons move freely within that lattice. Plastic ice VII, which is another hybrid phase and considered a precursor to the superionic state[11,18], is predicted to exist at less extreme conditions along the ice VII melting line[7–11].

Plastic crystal phases, intermediate between liquids and crystals and with unique mechanical properties such as pronounced plastic flow, high compressibility and reduced thermal conductivity[13,14,25,26], have been identified in molecular solids[27,28], water mixtures[29] and ionic solids[13,14,25,26]. Whether such phases can also be present in water ice, where strongly directional hydrogen bonds typically promote well-defined crystalline structures, remains uncertain. The observation of a decoupling of translational and rotational degrees of freedom in strongly confined high-density water on freezing suggests a high-density amorphous plastic phase may have formed[30]. Yet the continuous slope of the ice VII melting curve (determined through direct visualization of sample melting[31]) disagrees with classical molecular dynamics simulations that have found discontinuous melting due to the emergence of the plastic phase[7,8,10]. Whereas such disagreement raises doubts about the existence of a plastic ice VII phase, ab initio molecular dynamics simulations have also identified dynamically disordered states near the ice VII melting line, and suggest a continuous transition to a plastic phase[11,18]. We also note that the lack of direct experimental evidence for a plastic ice phase reflects the challenge of observing a phase transition that primarily involves changes in hydrogen dynamics rather than structural transformations.

[1]Dipartimento di Fisica, Sapienza Università di Roma, Roma, Italy. [2]Laboratory of Quantum Magnetism, Institute of Physics, École Polytechnique Fédérale de Lausanne, Lausanne, Switzerland. [3]School of Mathematics, University of Bristol, Bristol, UK. [4]IBM Research Europe, Daresbury, UK. [5]Centre for Science at Extreme Conditions and School of Physics and Astronomy, University of Edinburgh, Edinburgh, UK. [6]Institut de Minéralogie, de Physique des Matériaux et de Cosmochimie (IMPMC), CNRS UMR7590, Sorbonne Université, Paris, France. [7]Institut Laue-Langevin (ILL), Grenoble, France. [8]Department of Chemical Engineering, The University of Manchester, Manchester, UK. [9]Laboratoire Leon Brillouin, CNRS-CEA, Saclay, France. [10]Present address: Centro de Física de Materiales (CFM-MPC), CSIC-UPV/EHU, Donostia/San Sebastián, Spain. ✉e-mail: livia.bove@upmc.fr

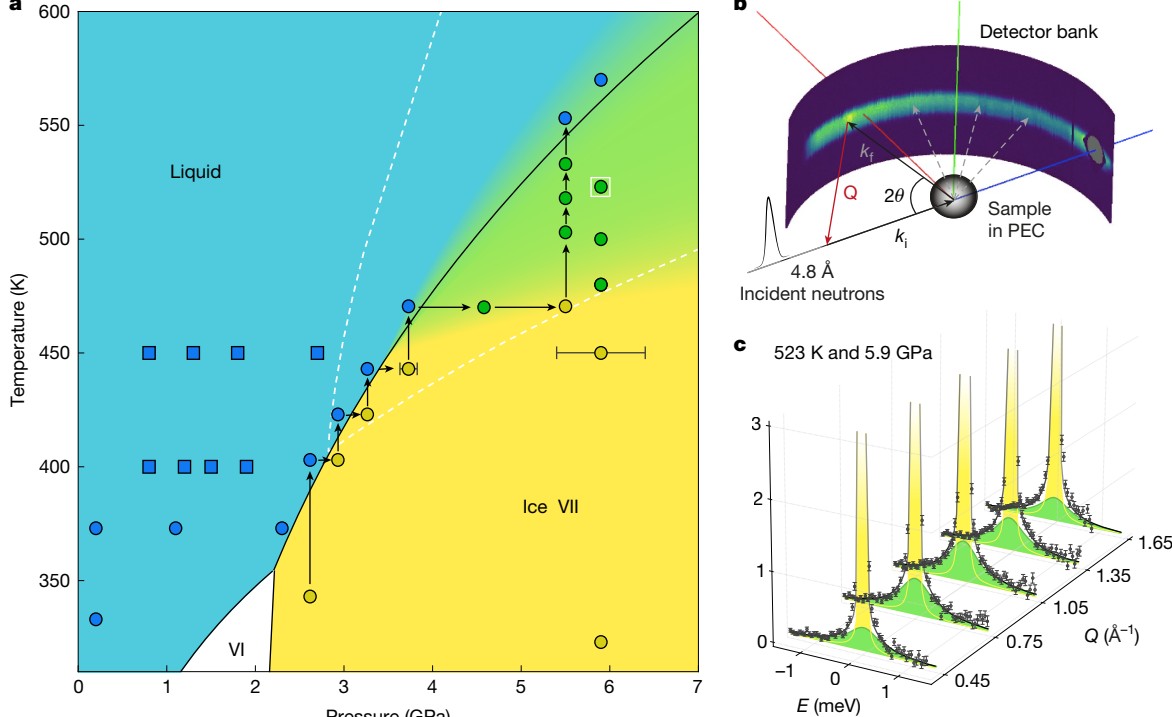

**Fig. 1 | Phase diagram of the QENS experiment and investigated thermodynamic paths. a**, Experimental phase diagram of water in the investigated *P–T* region, melting lines are reproduced from refs. 31,43. The region of stability of plastic ice VII from numerical simulations[8] is shown with white dashed lines, pressures and temperatures are shifted to match the experimental melting line. The measured thermodynamic points are represented as circles with colours corresponding to the phase attribution (a selection of thermodynamic points from previous experiments[34] are represented as squares). Error bars on the pressure determination are of ±0.5 GPa for the high-pressure isobars, whereas they are of 0.1 GPa for the

thermodynamic points close to the melting as the melting was used to calibrate the pressure. **b**, Graphical sketch of a QENS experiment. The detector bank shows raw data from a real QENS experiment on IN5, ILL, Grenoble (France). $k_i$ and $k_f$ represent the initial and final wavevectors of the scattered neutrons; $2\theta$ is the scattering angle; and Q is the exchanged wavevector. **c**, Q slices of the measured $S(Q,\omega)$ in plastic ice VII at 523 K and 5.9 GPa, the elastic peak is represented as a yellow area whereas the rotational contribution is represented as a green area. Error bars were calculated by the square root of absolute neutron count combined with the law of propagation of errors.

Our approach to this problem is to probe the dynamics of water molecules along the high-temperature melting curve of ice VII within the temperature range from 450 to 600 K through quasi-elastic neutron scattering (QENS) experiments at the time-of-flight (TOF) spectrometers IN5 and IN6-SHARP of the Institut Laue-Langevin (ILL), using a Paris–Edinburgh press device[32] to generate pressures between 3 and 6 GPa. The measured *P–T* paths are shown in the experimental phase diagram of water, in Fig. 1a. QENS probes stochastic, both diffusive and rotational, motions in hydrogenated materials. The scattering process enabling this is sketched in Fig. 1b, with the high incoherent neutron scattering cross-section of hydrogen (roughly 80 barn, compared to roughly 2 barn for deuterium) ensuring that the measured scattering primarily captures the self motion of the hydrogen atoms in the system[33–35]. The scattered neutron beam is detected and analysed as a function of both the energy transfer $\omega$ and wave vector transfer $Q$ to measure the dynamical structure factor $S(Q,\omega)$ of the system (an example is reported in Fig. 1c). As indicated by the term 'quasi-elastic', QENS primarily investigates small energy transfers, that is, motions occurring on a time scale comparable to the resolution of the instrument that in this study is typically in the range of picoseconds. Diffusive and rotational motions can be distinguished by analysing the QENS signal broadening and intensity profile as a function of $Q$ (refer to the Methods section for full details). Figure 2 shows examples of the measured $S(Q,\omega)$ signal (Fig. 2a–c) and of its projection for a selected $Q$ value (Fig. 2d–f) in the liquid, plastic ice and ordinary ice VII, respectively (Fig. 2g–i sketches their structures).

Crossing the ice VII melting line along isotherms and isobars (Fig. 1a), we identified a region in the phase diagram in which water molecules

show liquid-like orientational dynamics on the picosecond time scale while preserving the crystalline ice structure as determined by X-ray and neutron diffraction (see Methods for details). The broadening and intensity of the QENS signal with respect to the exchanged wave vector ($Q$) clearly differentiates between (1) the liquid phase, characterized by global $Q$-dependent quasi-elastic broadening and by the presence of both translational and rotational components (Fig. 2d and Extended Data Fig. 1a), (2) the plastic crystal, in which translational diffusion is arrested while rotational dynamics persist (Fig. 2e and Extended Data Fig. 1b) and (3) the ordinary ice VII crystal, in which both translational and rotational dynamics are frozen on the time scale of the instrument, resulting in no quasi-elastic broadening of the elastic line (Fig. 2f and Extended Data Fig. 1c). In the plastic crystal region, data analysis with a single Lorentzian contribution shows a quasi-elastic peak width that remains relatively constant with the exchanged wave vector, indicating localized motion of the water molecules, such as reorientations (Extended Data Figs. 1b and 2a). Compared to the liquid and as predicted, the plastic crystal thus represents a unique intermediate state in which molecules have lost the ability to translate freely and are arranged in an ordered crystalline structure but retain their capacity to rotate. Seven thermodynamic points corresponding to plastic ice VII have been measured, marked as green dots in Fig. 1a.

QENS can also provide insight into the specific rotational mechanisms by the quasi-elastic contribution's intensity evolution with $Q$. Analysis of the $Q$ dependency of the rotational component and a direct comparison of measured rotational time scales and intermediate scattering functions with molecular dynamics simulations (see Methods

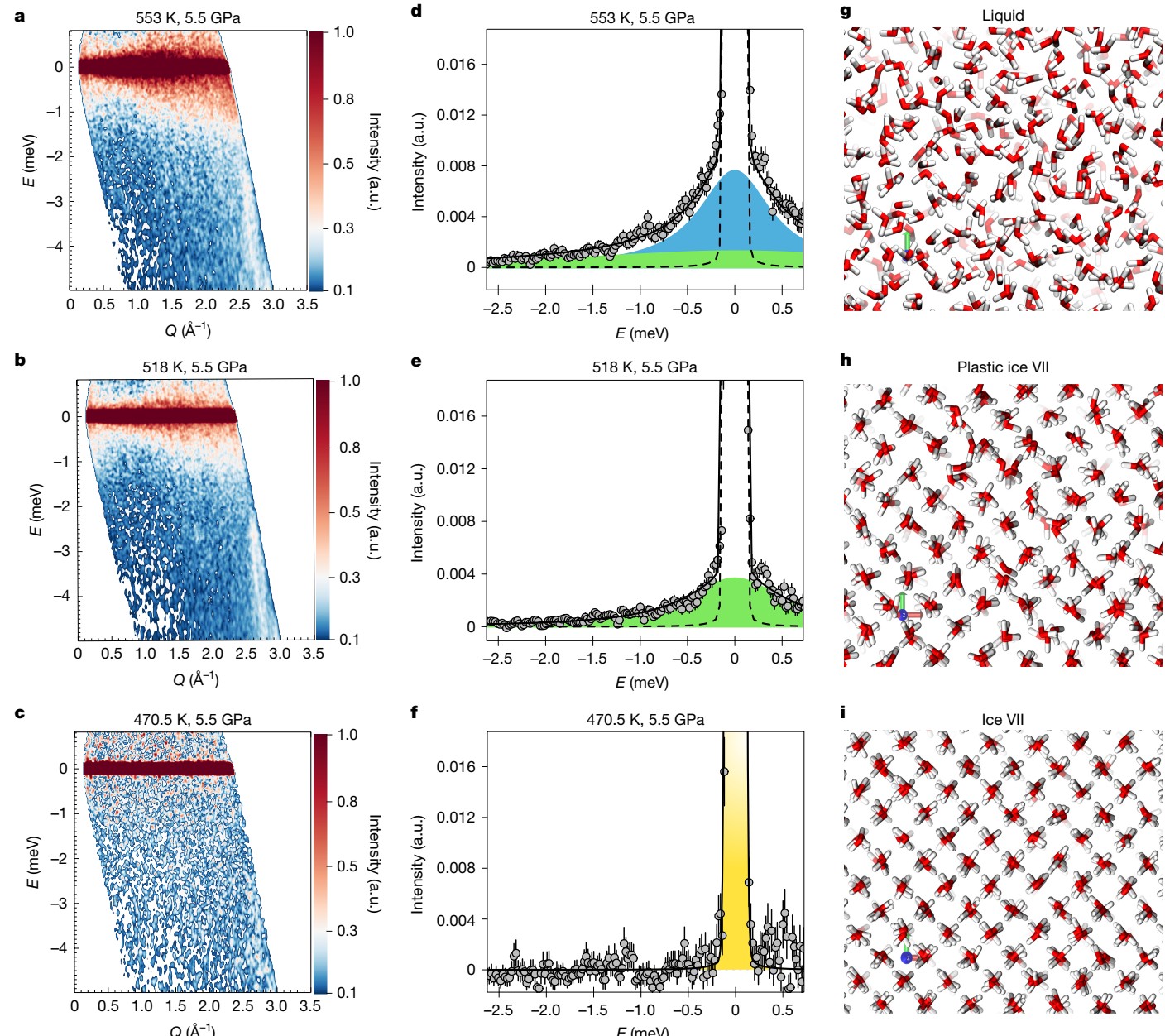

**Fig. 2 | Experimental data and molecular dynamics snapshots in the three phases. a**–**c**, The measured $S(Q,\omega)$ at 5.5 GPa in the liquid phase (553 K) (**a**), plastic ice VII (518 K) (**b**) and ordinary ice VII (470.5 K) (**c**). **d**–**f**, QENS spectra at a fixed $Q$ value of 1.2 Å$^{-1}$ in the same $P$–$T$ conditions: liquid phase (**d**), plastic ice VII (**e**) and ordinary ice VII (**f**). The black dashed line represents the instrumental resolution, the green area represents the rotational quasi-elastic Lorentzian and the blue area represents the translational Lorentzian. Error bars were calculated by the square root of absolute neutron count combined with the law of propagation of errors. **g**–**i**, Snapshots from our molecular dynamics simulations in the liquid (**g**), plastic (**h**) and ordinary ice VII (**i**) phases. a.u., arbitrary units.

for details) revealed that plastic ice VII does not conform to a free rotor phase as seen in methane and ammonia ices. Instead, water molecule orientations change randomly between preferential directions dictated by the crystal field and separated by potential energy barriers that are sufficiently small to allow molecules to switch rapidly, on a picosecond time scale, from one orientation to another. Such swift orientational jumps have been previously observed in other tetrahedral plastic crystals[13], which are not categorized as free rotors.

The tetrahedral symmetry of the water molecule and the lattice structure of ice VII allow for four types of rotation. With hydrogen sites half-occupied, the water molecule can rotate around the three twofold ($C_2$) and four threefold ($C_3$) axes of the tetrahedron (Extended Data Fig. 3c). $C_2$ rotations shift the positions of all hydrogen atoms; in

$C_3$ rotations, three hydrogen atoms exchange places while the fourth on-axis atom, remains stationary. The quasi-elastic scattering function for rotations around both axes is identical (see equation (8) in Methods). Given the lattice structure of ice VII, two conformations exist for the water molecule's tetrahedron, with the eight hydrogen sites located at the cube's vertices. The molecule can either reorient around the $C_2$ or $C_3$ axes as described above, or perform a fourfold rotation around the cubic axis ($C_4$) (cubic tumbling), allowing hopping among the cube's eight corners (Fig. 3c). The fourth possibility is isotropic rotation in which hydrogen atoms visit all possible sites on the surface of a sphere (equation (6)). We fitted our plastic ice VII data using these reorientational models (Fig. 4 and Extended Data Figs. 3, 4; see Methods for details), but the limited $Q$ range and the inability to

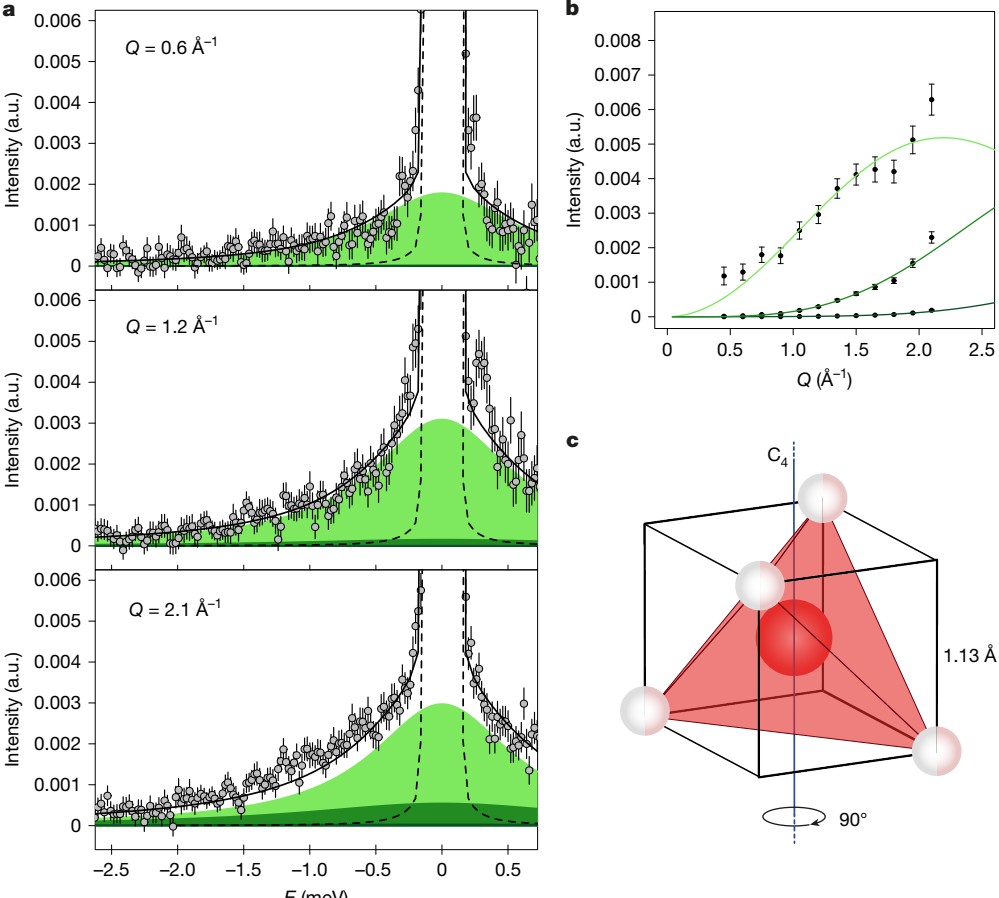

**Fig. 3 | C₄ rotational model. a**, Examples of fits to the data of plastic ice VII ($P$, 5.5 GPa; $T$, 518 K) using the three Lorentzians in equations (9), (10) and (11), coloured areas represent the fitted Lorentzians. The relative intensities and widths of the three contributions are constrained to the model, the intensity factor at each $Q$ value is left as a free parameter. Error bars were calculated by the square root of absolute neutron count combined with the law of propagation of errors. **b**, Intensity of the three quasi-elastic Lorentzians as a function of $Q$, obtained as described in the 'Fitting procedure' section in the Methods. For both **a** and **b**, light green refers to the most intense Lorentzian (equation (11)), green represents equation (10) and dark green is equation (9). Error bars represent standard deviations of the fitted parameters. **c**, Geometrical representation of one of the C₄ axes of the cube around which the molecule reorients.

normalize quasi-elastic intensities—due to the strong contribution from the high-pressure apparatus—prevent us from distinguishing between the models as they primarily differ above 3 Å⁻¹ and in the predicted intensities. Nonetheless, analysis of the fitting residuals indicated that the isotropic rotation model was more poorly describing our data. Molecular dynamics simulations were performed to investigate the reorientational mechanisms further (see Fig. 2g–i for simulation snapshots of the three phases). The computed orientational probability density function of ordinary ice VII shows sharp and well-defined maxima (Fig. 4a). In plastic ice VII, the same preferred molecular orientations can be recognized, with a broader distribution, confirming that the rotations are not isotropic. A Markov chain analysis from the simulations indicates that the most frequent rotational transition of hydrogen atoms corresponds to C₄ rotations (more details are given in the Methods); hence, our QENS data were analysed using the fourfold rotational model.

Using the procedure outlined in the Methods, we obtained estimates for the reorientational time $\tau_{90°}$, the hydrogen mean vibrational amplitude $\langle u^2 \rangle$ and the intensity of each Lorentzian (presented as a function of $Q$ in Fig. 3b, alongside the model function). For plastic ice VII at 518 K and 5.5 GPa, we find a reorientational time $\tau_{90°}$ of 0.51 ± 0.01 ps. Table 1 lists $\tau_{90°}$ estimates for all investigated thermodynamic points and compares them to estimates from molecular dynamics simulations carried out under similar conditions. Simulated reorientational times are estimated

from fits of the rotational auto-correlation function with equation (5) in Supplementary Information. Examples of fits to the intermediate scattering function are shown in Fig. 4b. Hydrogen mean vibrational amplitude results are also presented in Table 1 for all measured state points, showing that in the plastic phase $\langle u^2 \rangle \simeq 0.80 \pm 0.1$ Å², that is, roughly four times higher than in the ordinary crystal. These vibrational amplitudes are compatible with simulated results using the TIP4P/2005 water model (obtained from the plateau of the hydrogen's mean square displacement), also reported in Table 1, where $\langle u^2 \rangle \simeq 1.15 \pm 0.01$ Å² for all state points.

The reorientational mechanism observed in plastic ice VII resembles the reorientation through rapid molecular jumps reported for liquid water[36,37]; our measured reorientational times of 0.5–0.7 ps are found to be comparable to what was measured in liquid water in the same temperature range[33,34,38]. Our Markov chain analysis also shows that rotations arise from variations in coordination patterns and involve cooperative molecular motions as in liquid water[39], but reorientations in plastic ice VII are restricted by the local crystal environment and only allow jumps between high-symmetry directions and some transient intermediate positions. This also explains the persistence of the (111) reflection of our neutron diffraction patterns (see 'Neutron diffraction experiment' section in the Methods and Extended Data Fig. 5), which indicates a preferential orientation of the hydrogens along the diagonals of the cube.

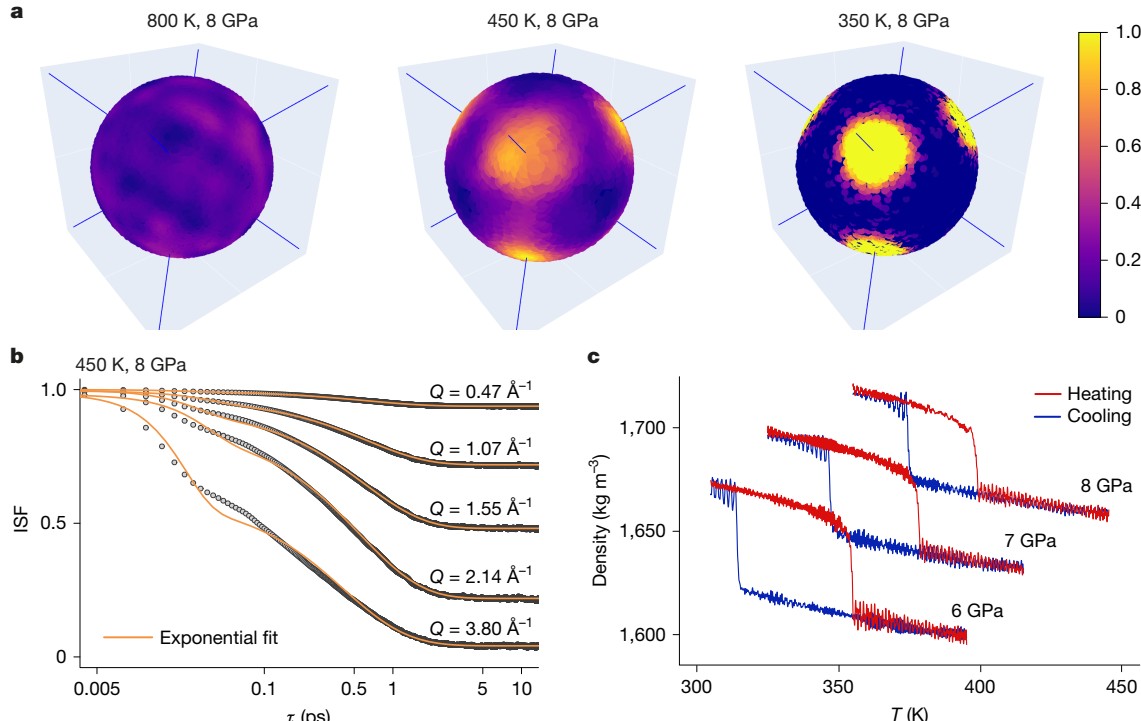

**Fig. 4 | Molecular dynamics simulations. a**, Positions of the hydrogen atoms obtained from molecular dynamics simulations in the liquid (8 GPa and 800 K), plastic (8 GPa and 450 K) and ordinary solid (8 GPa and 350 K) phase. **b**, Rotational intermediate scattering functions in the plastic phase (8 GPa and 450 K) as obtained from molecular dynamics simulations, orange lines are fits obtained with equation (5) in the Supplementary Information. **c**, Density as a function of temperature for three pressures (6, 7, 8 GPa) during heating and cooling cycles. The heating and cooling rate is 5 K per 100 ps.

The rotational dynamics of plastic ice VII is associated with significant sublattice mixing, differentiating it from the dynamics of ordinary ice VII (ref. 40). Dielectric experiments[41] probing room temperature dynamics of ice VII, observed transitions from molecular rotation to proton hopping and then to proton tunnelling with increasing pressure, reflecting the dynamics of the high-temperature plastic and superionic phases. Our finding that the rotational dynamics in plastic ice VII is marked by an almost fourfold increase in hydrogen vibrational amplitude highlights the interplay between rotation and structure. Molecular orientations in liquid water correlate with H-bond network distortions[39], a role assumed in plastic ice VII by lattice distortions and explaining the relative independence of the reorientational time from temperature and pressure (Table 1). The density variation when transitioning from plastic ice VII to ice VII when simulating heating and cooling cycles at different pressures (Fig. 4c) corresponds to a 2% volume change, which suggests a first-order phase transition, as found in previous classical simulations[7–9]. This prompted in-laboratory X-ray diffraction (XRD) experiments (details are given in the Methods), observing a volume jump across the crystal to plastic transition compatible with the numerical simulations (Extended Data Fig. 6). However, the discontinuous volume change was only observed at a single point in the plastic state because the XRD signal was disrupted by single crystal formation when approaching the ice VII melting line. A more focused beam and shorter measuring time, probably at synchrotron facilities, are required to confirm whether the ice VII to plastic ice VII transition is accompanied by a volume discontinuity and hence whether it is a first-order or continuous transition.

In conclusion, we directly observed plastic ice VII along its high-temperature melting curve using QENS measurements and molecular dynamics simulations and showed it to be a jump-rotor plastic crystal. The jump-rotor nature of plastic ice suggests enhanced thermal conductivity and elastic properties compared to the predicted free rotor crystal, but reduced with respect to ordinary ice VII. Other materials with tetrahedral coordination have recently been shown[13,14] to form rotor plastic phases under high pressure, with low fusion enthalpies of around 5–6 kJ mol[-1] as measured for high-temperature ice VII (ref. 42), higher compressibility and showing significant barocaloric effects that may also be present in plastic ice VII. These properties are

## Table 1 | Reorientational times $\tau_{90°}$ and hydrogen mean square vibrational amplitudes $\langle u^2 \rangle$

| Experiment | | | | Simulations | | | |
|---|---|---|---|---|---|---|---|
| $T$ (K) | $P$ (GPa) (±0.5) | $\tau_{90°}$ (ps) | $\langle u^2 \rangle$ (Å²) | $T$ (K) | $P$ (GPa) | $\tau_{90°}$ (ps) | $\langle u^2 \rangle$ (Å²) |
| 470.5 | 4.6 | 0.56 (0.02) | 0.90 (0.52) | 450 | 6 | 0.38 (0.01) | 1.14 (0.01) |
| 518 | 5.5 | 0.51 (0.01) | 0.78 (0.02) | 500 | 7 | 0.32 (0.02) | 1.16 (0.01) |
| 533 | 5.5 | 0.53 (0.01) | 0.83 (0.03) | 550 | 7 | 0.26 (0.02) | 1.15 (0.01) |
| 480 | 5.9 | 0.70 (0.03) | 0.85 (0.06) | 450 | 8 | 0.46 (0.04) | 1.10 (0.01) |
| 523 | 5.9 | 0.67 (0.02) | 0.72 (0.09) | 500 | 8 | 0.34 (0.02) | 1.12 (0.01) |

In this table we compare the $\tau_{90°}$ and $\langle u^2 \rangle$ parameters as found in the plastic ice VII from QENS experiments and in our prediction from molecular dynamics simulations. As in the experimental data the main contribution to the scattering is from the HDO molecules, the experimental reorientational times are rescaled by the different moment of inertia of HDO and $H_2O$ molecules, for a better comparison with molecular dynamics simulations performed on fully hydrogenated molecules. Uncertainties are given in parenthesis.

relevant to icy bodies in our solar system, and their quantification is important for developing more accurate models of such bodies' internal dynamics and glacial flow. The presence of a plastic phase of ice VII may have played a key role in the differentiation processes of large icy moons, potentially explaining the differences in differentiation states observed between Ganymede, Callisto and Titan[2]. In closing, we note that plastic phases typically involve simple, symmetric molecules such as methane or alkanes, which makes the discovery of a plastic state in ice VII notable in view of water's bent geometry and hydrogen bonding. The hydrogen bonds persist during the plastic transition and enable cohesive interactions, and such behaviour might be found in a larger class of hydrogen-bonded materials under high-pressure and high-temperature conditions.

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

## Methods

### QENS experiments

The experiments were performed on IN6-SHARP[44] and IN5 (ref. 45). The incident wavelength on IN6-SHARP was 5.1 Å and on IN5 it was 4.8 Å. With this incoming beam wavelength, the accessible $Q$ range was between 0.45 and 1.65 Å$^{-1}$ on IN6-SHARP, and between 0.45 and 2.1 Å$^{-1}$ on IN5. The sample was a mixture of $H_2O$ and $D_2O$ at a ratio of 10:90 to ensure a good transmission (50%). The use of a Paris–Edinburgh press with an opposed-anvil geometry designed for high temperatures allowed us to reach pressures up to about 6 GPa and temperatures close to 600 K (ref. 32). The sample was loaded in Inconel hemispherical gaskets and the total sample volume was about 60 mm$^3$. Temperature was measured by two thermocouples attached to each anvil with a precision of ±5 K, whereas pressure was determined by the force on the gasket and the melting curve of water reported in ref. 31. In Fig. 1b the IN5 detector bank with raw data is shown. It is worth mentioning why only the central portion of the detector bank is illuminated. In a Paris–Edinburgh press, the anvils are shielded with cadmium to limit the background contribution. This implies that the portion of detector bank that can be effectively used is limited by the angular aperture of the anvils, that is, neutrons that would be detected by the higher and lower portion of the detector bank are absorbed by the cadmium on the anvils. As pressure is increased and the anvils become progressively closer, the illuminated portion of the detector bank decreases as well as the total quasi-elastic intensity, limiting the maximum pressures achievable with high-pressure QENS. The measured thermodynamic points are shown in Fig. 1a. Data reduction was performed using the Large Array Manipulation Package, an interactive data language (IDL)-based data reduction package written at ILL for data analysis and manipulation[46]. Vanadium was used to normalize the data and to evaluate the experimental resolution and the relative detector efficiency[47]. The background in the TOF spectra was evaluated by integrating the intensity over a group of TOF channels far from the elastic and quasi-elastic regions or any other sample contribution. The background was found to be constant in TOF but had a non-negligible dependence on the scattering angle. This contribution was then subtracted before TOF to energy conversion. As a final step, data were binned in constant-$Q$ spectra with 0.15-Å$^{-1}$ steps. Multiple scattering corrections were also applied to the data, details are given in the Supplementary Information file (Supplementary Figs. 1–4).

Reduced QENS data contain information on the incoherent response of the sample, but they also contain an elastic contribution due to the scattering from the gasket and anvils' materials, both convoluted with the experimental resolution. As the gasket contribution is totally elastic but variable with pressure, because of the deformation of the gasket and the narrowing of each anvil's distance with an increase in loading, we preferred to model it directly into the data with an elastic contribution, rather than subtracting an empty cell contribution from the data before fitting. To properly account for the incoherent dynamical structure factor $S_{inc}(Q, \omega)$ of the sample, for the cell-scattering contribution $\delta(\omega)$, and for the instrumental resolution $R(Q, \omega)$, the fitting function $F(Q, \omega)$ can be written as:

$$F(Q, \omega) = [S_{inc}(Q, \omega) + \delta(\omega)] \otimes R(Q, \omega) + B \qquad (1)$$

where $B$ is a flat background that takes into account the sample dynamics that is too fast to be observed on the time scale of the experiment. In our analysis, because the resolution was sharp, we fitted the data using a non-convoluted incoherent dynamical structure factor and a pseudo-Voigt function taking into account the convolution of the elastic peak $\delta(\omega)$ with the resolution. In the following analysis, the half-width at half-maximum (HWHM) and the Gaussian–Lorentzian ratio of the pseudo-Voigt function were fixed to the parameters estimated from fits to the data in the crystalline phase, whereas its total intensity

was left as a free parameter. The incoherent dynamical structure factor $S_{inc}(Q, \omega)$ describes three potential contributions: the diffusive dynamics of the molecular centres of mass, the orientational dynamics of the molecules, and the vibrational dynamics of the molecules. To derive an analytical expression for $S_{inc}(Q, \omega)$, it is essential to assume the decoupling of these three contributions[33,34] (refer to the 'Decoupling spproximation' section in the Supplementary Information for more details on the validity of this assumption). Under this assumption, the incoherent dynamical structure factor can be expressed as the convolution of translational $S_T(Q, E)$, rotational $S_R(Q, E)$ and vibrational term $S_V(Q, E)$:

$$S_{inc}(Q, \omega) = S_V(Q, \omega) \otimes S_T(Q, \omega) \otimes S_R(Q, \omega). \qquad (2)$$

These three contributions show qualitatively distinct behaviour when studying their evolution as a function of the exchanged wave vector, enabling a clear distinction of the motions contributing to the measured QENS intensities.

**Vibrational contribution.** The vibrational contribution to the quasi-elastic spectra can be modelled as a damping factor. This multiplicative factor, the Debye–Waller factor, is expressed as $e^{-\langle u^2 \rangle Q^2/3}$, where $\langle u^2 \rangle$ is the hydrogen mean vibrational amplitude from its equilibrium lattice position. The incoherent dynamical structure factor is then expressed as the convolution of the translational and rotational contributions multiplied by the Debye–Waller factor:

$$S_{inc}(Q, \omega) = e^{-\frac{\langle u^2 \rangle Q^2}{3}} [S_T(Q, \omega) \otimes S_R(Q, \omega)] \qquad (3)$$

**Translational contribution.** The translational dynamical structure factor $S_T(Q, \omega)$ can be written as:

$$S_T(Q, \omega) = \frac{1}{\pi} \frac{\Gamma_T(Q)}{\omega^2 + \Gamma_T(Q)^2} \qquad (4)$$

with $\Gamma_T(Q) = D_T Q^2$ in the case of free diffusion or:

$$\Gamma_T(Q) = \frac{D_T Q^2}{1 + (Q \times d)^2/6} \qquad (5)$$

for jump diffusion[48], as is the case for water[33,34], where $D_T$ is the translational diffusion coefficient and $d$ is the apparent jump distance.

**Rotational contribution.** In the simple case of isotropic rotations, as holds for liquid water at high $P$ and $T$ (refs. 33,34), the rotational dynamical structure factor is given by the Sears expansion[49]:

$$S_R(Q, \omega) = j_0^2(aQ)\delta(\omega) + \sum_{l=1}^{\infty} (2l+1)\frac{j_l^2(aQ)}{\pi} \frac{D_R l(l+1)}{[D_R l(l+1)]^2 + \omega^2} \qquad (6)$$

where $D_R$ is the rotational diffusion coefficient, $j_l(x)$ are spherical Bessel functions and $a = 0.98$ Å is the OH distance in the water molecule[50].

For non-isotropic rotations, several models are described in the literature[50]. We focus here on the two most plausible cases given the crystalline geometry of ice VII, that is, cubic tumbling and reorientations around the $C_2$ or $C_3$ symmetry axes of the cubic crystal. For all models of discrete molecular reorientations, the scattering function is given by:

$$S_R(Q, \omega) = A_0(Q)\delta(\omega) + \sum_{i > 0} \frac{A_i(Q)}{\pi} \frac{1/\tau_i}{(1/\tau_i)^2 + \omega^2} \qquad (7)$$

with $\sum_{i>0} A_i(Q) = 1 - A_0(Q)$. In the case of reorientations around the $C_2$ or $C_3$ symmetry axis, only one Lorentzian contributes and its intensity is[51]:

$$A_1(Q) = \frac{1}{2}[1 - j_0(d_{HH}Q)] \qquad (8)$$

where $d_{HH} = (2\sqrt{2}/\sqrt{3})a$ is the average distance between the two hydrogens in the water molecule, equal to 1.60 Å, and $j_0(x) = \sin(x)/x$ is the zeroth-order spherical Bessel function.

In the cubic tumbling model instead, reorientations occur as 90° jumps around the three $C_4$ symmetry axis of the cube in which the water molecule tetrahedron is embedded (Fig. 3). In this case, as described in ref. 52, the intensities of the three Lorentzians contributing to $S_R(Q, \omega)$ are:

$$A_1(Q) = \frac{1}{8}[1 - 3j_0(dQ) + 3j_0(\sqrt{2}\,dQ) - j_0(\sqrt{3}\,dQ)] \qquad (9)$$

$$A_2(Q) = \frac{1}{8}[3 - 3j_0(dQ) - 3j_0(\sqrt{2}\,dQ) + 3j_0(\sqrt{3}\,dQ)] \qquad (10)$$

$$A_3(Q) = \frac{1}{8}[3 + 3j_0(dQ) - 3j_0(\sqrt{2}\,dQ) - 3j_0(\sqrt{3}\,dQ)] \qquad (11)$$

where $d = (2/\sqrt{3})a$ is the edge of the cube where the tetrahedron is embedded, equal to 1.13 Å. The $\tau_i$s are related to the mean residence time between two successive reorientational jumps $\tau_{90°}$ by:

$$\frac{1}{\tau_1} = \frac{2}{\tau_{90°}}, \qquad \frac{1}{\tau_2} = \frac{4}{3\tau_{90°}}, \qquad \frac{1}{\tau_3} = \frac{2}{3\tau_{90°}} \qquad (12)$$

**Fitting procedure.** We started at the low-temperature points to have a reference for data analysis. Fitting data in the ordinary solid phase gave us a benchmark to describe the experimental resolution in the following analysis. In the ice VII case, as discussed above, there is no quasi-elastic intensity and the signal is simply given by the elastic Dirac delta function $\delta(\omega)$ convoluted with the experimental resolution (a pseudo-Voigt function). In addition to the high-pressure cell contribution, the Dirac delta function is also due to the elastic scattering contributions from the sample (long-term lattice structure and dynamics slower than the instrumental resolution). Examples of fits in the ordinary ice VII phase are given in Extended Data Fig. 1a in which the yellow area represents the pseudo-Voigt resolution function.

Second, for the liquid phase, we modelled $S_{inc}(Q, \omega)$ as the convolution of the vibrational, translational and rotational terms, using the isotropic model for the rotational contribution and the first term of the sum in equation (6) only (as this is the dominant term over the probed $Q$ range). Then equation (3) becomes:

$$S_{inc}(Q, \omega) = e^{-\frac{\langle u^2 \rangle Q^2}{3}}$$
$$\left[ \frac{j_0^2(aQ)}{\pi} \frac{\Gamma_T(Q)}{\omega^2 + \Gamma_T(Q)^2} + \frac{3j_1^2(aQ)}{\pi} \frac{2D_R + \Gamma_T(Q)}{\omega^2 + [2D_R + \Gamma_T(Q)]^2} \right] \qquad (13)$$

where $a = 0.98$ Å is the OH distance in the water molecule and $\Gamma_T(Q)$ is the HWHM for the jump diffusion model expressed as in equation (5). Spectra at all $Q$ values were fitted simultaneously to impose the ratio of the intensity of the two Lorentzian contributions and the behaviour with $Q$ for the HWHM of the two contributions. With this procedure, the free parameters of the fit are the total intensity factor at each $Q$ value, and the rotational and translational diffusion coefficients. To reduce the number of free fitting parameters, we first constrain the rotational diffusion coefficient to the value extrapolated from the literature[34]. We then constrain the value obtained for the translational

diffusion coefficient and treat the rotational diffusion coefficient as free parameter. With this procedure we obtain for the translational diffusion coefficient $(8.0 \pm 0.8)10^{-5}$ cm$^2$ s$^{-1}$ and for the rotational diffusion coefficient $(1.1 \pm 0.3)$ ps$^{-1}$ at 553 K and 5.5 GPa. Examples of fit in the liquid phase are shown in Extended Data Fig. 1c, in which the black dashed line represents the instrumental resolution, and the blue and green areas represent the translational and rotational Lorentzian, respectively. The HWHMs of the two contributions are shown in Extended Data Fig. 2b.

As for the plastic ice VII, in the absence of translational diffusion, the only contributions to the incoherent dynamical structure factor are the vibrational and rotational terms:

$$S_{inc}(Q, \omega) = e^{-\frac{\langle u^2 \rangle Q^2}{3}} S_R(Q, \omega) \qquad (14)$$

As a first step, we focused our analysis on the data at 518 K and 5.5 GPa. By fitting these data with the sum of one single Lorentzian with free parameters and a pseudo-Voigt function, we obtain an HWHM for the quasi-elastic Lorentzian, which is constant in $Q$, which indicates localized motions. HWHMs obtained with this procedure are shown in Extended Data Fig. 2a. Examples of fit at 518 K and 5.5 GPa with one Lorentzian are shown in Extended Data Fig. 1b, in which the black dashed line represents the instrumental resolution, and the green areas represent the rotational Lorentzian. To choose a reorientational model to describe our quasi-elastic intensities, the spectra at each $Q$ value are fitted simultaneously, constraining the widths of the Lorentzians to be constant in $Q$ and the intensities to the specific rotational model considered, while leaving the value of the reorientational time (or equivalently, the HWHM) and an intensity factor as free parameters. The hydrogen mean vibrational amplitude $\langle u^2 \rangle$ is constrained to the value extrapolated from literature data[53], of 0.8 Å$^2$. Examples of fits at three $Q$ values are shown in Fig. 3a and Extended Data Figs. 3a, 4a. We can then release the constraint on the $Q$ behaviour of the intensities, still constraining the relative intensities of the Lorentzian contributions. Results of the behaviour of the intensities are shown in Fig. 3b and Extended Data Figs. 3b, 4b. The fit quality, both constraining the intensity behaviour with $Q$ and relaxing the constraints on the intensities, is comparable for the three models (isotropic, reorientation around $C_2/C_3$ axes and cubic tumbling). Nevertheless, analysis of residuals indicates that the fit quality when constraining the fit to the isotropic rotational model is slightly worst. We then relied on the molecular dynamics simulations for the choice of the non-isotropic rotational model to use.

In the final analysis, data are fitted with the $C_4$ reorientational model (equations (9)–(11)), constraining the ratios of the widths of the three Lorentzians and their relative intensities, and leaving the intensity of the most intense contribution at each $Q$ value, the reorientational time and $\langle u^2 \rangle$ as free parameters. Results of this fitting procedure at each investigated thermodynamic condition are reported in Table 1 and Fig. 3.

### Neutron diffraction experiment

Neutron diffraction experiments were performed at the D20 beamline in ILL, Grenoble (France)[54]. The high-pressure high-temperature setup was the same as for the QENS experiments, described above and using the gasket/anvil setup described in ref. 55. The incident neutron wavelength was $\lambda = 1.54$ Å.

The sample was D$_2$O loaded with MgO powder to avoid single crystal formation. MgO reacted with water forming Mg(OD)$_2$, nevertheless the diffraction pattern of ice was unaffected. As the sample is compressed and heated in the region where we observe an active rotational dynamics (1a), the (111) reflection of ice VII does not disappear.

An example of diffractogram at 500 K and 4.7 ($\pm 0.5$) GPa is reported in Extended Data Fig. 5.

## XRD experiments

XRD experiments were performed on the powder diffractometer at IMPMC, Paris (France). The X-ray source was a MicroMax-007 HF form Rigaku with Mo K$\alpha$ anode ($\lambda$ = 0.7107 Å). The beam size was about 100 μm. The diffractometer was equipped with a R-AXIS IV++ detector from Rigaku. Pressure was generated with a membrane Diamond Anvil Cell, diamonds had a culet of 500 μm. The rhenium gasket was indented to an initial thickness of about 50 μm. The hole in the gasket was laser drilled with a diameter of 200 μm and covered with a gold ring with an internal diameter of 160 μm (the thickness of the ring was 40 μm) to prevent reaction of the sample with the gasket and for pressure calibration.

Liquid milli-Q water was loaded in the cell, pressure was increased by inflating the membrane up to the crystalization pressure of ice VI and further compressed to obtain ice VII. The membrane Diamond Anvil Cell was externally heated, temperature was measured with a thermocouple in contact with the diamonds. Pressure was measured using gold as internal calibrant (using the equation of state from ref. 56). Powder diffraction data were acquired along two isobars (5.7 and 6.5 GPa), varying the temperature from 300 up to 563 K. The average acquisition time was around 30 min.

The acquired diffraction patterns were refined (Le Bail refinement) to extract volumes for ice VII and gold. Unit cell volumes were corrected to obtain ideal isobars, by correcting the data for the volume change associated to pressure variation (d$V/V$ = d$P/B$, where $B$ is the extrapolated value of the bulk modulus of ice VII). Volumes as a function of temperature are shown in Extended Data Fig. 6.

These preliminary data show that at roughly 500 K and 5.7 GPa, and at roughly 550 K and 6.5 GPa we observed a departure from a linear trend of about 1–2%, consistent with the volume jump predicted from molecular dynamics simulations. Along both isobars, data acquisition was interrupted due to single crystal formation (and consequent poor data quality) at higher temperatures. Future investigations are required to confirm this result and better identify the transition line between ice VII and plastic ice VII.

## Molecular dynamics simulations

Numerical experiments were based on classical molecular dynamics simulations[57] of systems composed of $N$ = 8,192 rigid water molecules described by the TIP4P/2005 interaction potential[58] in the isobaric (N$p$T) ensemble. Classical molecular dynamics simulations were performed with the GROMACS v.2020.2 software package[59]. Equations of motions were integrated with the Verlet algorithm[60] with a time step of 1 fs. Coulombic and Lennard–Jones interactions were calculated with a cut-off distance of 1.1 nm and long-range electrostatic interactions were treated using the particle-mesh Ewald algorithm. Temperatures and pressures were controlled using a Nosé–Hoover thermostat[61,62] with the period of the kinetic energy oscillations between the system and the reservoir set to 50 ps, and a Parrinello–Rahman barostat[63] with a time constant of 100 ps.

The initial ice VII configuration at 0 K contained 8,192 water molecules and this was obtained using the GenIce software package[64,65]. An initial equilibration was performed for 5 ns along the isotherm at $T$ = 450 K and $P$ = 6, $P$ = 7 and $P$ = 8 GPa. The following production runs lasted 500 ps and were performed at the same thermodynamic conditions, saving coordinates every 4 fs. Each thermodynamic point was the result of the average of five independent simulations. Further details on the simulations are given in the Supplementary Information. In particular, Supplementary Figs. 5–8 give information on the fitting of the computed intermediate scattering functions, and Supplementary Fig. 9 shows a direct comparison of the measured and computed dynamical structure factors.

**Markov chain analysis.** In plastic ice VII, translational degrees of freedom are not available. Therefore, neglecting vibrational motions, the space explored by hydrogens is reduced to the surface of a sphere centred in the lattice positions occupied by each molecule. As reported in Fig. 4a, hydrogen's orientational probability distribution is not isotropic, and the regions with higher probabilities point towards the eight closest water neighbours. These regions are divided into two sublattices, each corresponding to one of the interpenetrating networks found in the ordinary ice VII crystal. As a first approximation of the fast dynamics of molecular rotations, we consider a Markov approach. In this approximation, each state visited at a given time does not have the memory of the state visited at previous times[10]. In our case, a state is an orientation of one hydrogen of the central water molecule in Extended Data Fig. 7a. We divide the rotational surface into octants, one for each preferential direction. Considering that two hydrogen atoms cannot simultaneously occupy the same octant, the possible distinct configurations amount to 28. Within this model, small oscillations around preferential directions (librations) do not influence the configurational state and are therefore neglected. We trained our model using 50-ps long trajectories at 8 GPa, and sampling the molecular orientation every 4 fs. The choice of 8 GPa is guided by the absence of positional exchange between water molecules, which instead occurs (although seldom) at lower pressures. From our data, we can define a transition matrix representing the transition probabilities between different states or orientations. On evolving the transition matrix to equilibrium, we can recognize three main groups of geometrically equivalent conformations with different probabilities of being occupied. The three states correspond to the following geometries: (1) stable (S) configurations, corresponding to configurations in which both hydrogen atoms are bonded to the same sublattice. They represent the lowest energy configurations and are the only configurations found in ice VII. (2) Edge (E) configurations, corresponding to configurations in which the two hydrogen atoms are oriented along the edges of the cubic lattice so that the central molecule is bonded to both sublattices. (3) Diagonal (D) configurations, corresponding to configurations in which the two hydrogen atoms are oriented along the main diagonal of the cubic lattice. As for the edge configuration, in this case, the central molecule is bonded to both sublattices.

Extended Data Fig. 7a reports a pictorial representation of the configurations mentioned above; the water molecules in the first shell of the central molecule belong to one of the two independent sublattices and are, therefore, represented with spheres of different colours (blue and red). The equilibrium probabilities relative to the three configurations are reported in Extended Data Fig. 7b and indicate that the stable state (being the energy-favoured state) is the most probable. To uncover the kinetics of molecular rotations in plastic ice VII, we consider all transition pathways between stable states. The transitions are sketched in Extended Data Fig. 7c and the relative probabilities are reported in Extended Data Fig. 7b. We can recognize two main groups of rotations: direct transitions between stable states (S → S) and indirect transitions that go through intermediate states (S → E → S and S → D → S, respectively). We label the transitions between stable states (either directly or through intermediate states) with the corresponding rigid rotation that connects the initial and final orientation of the molecule. $C_1$ are transitions that do not change the orientation of the molecule; $C_2$ are 180° rotations along the twofold axis of the cubic lattice; $C_3$ are 120° rotations along the threefold axis (main diagonal) of the cubic lattice and $C_4$ are 90° rotations along the fourfold axis of the cubic lattice. We furthermore observe that intermediate transitions from the stable state to intermediate states occur mostly by means of $C_6$ rotations along the threefold axis (main diagonal) of the cubic lattice, with only one of the hydrogen changing octant during the transition. This is just an idealized pathway, with the real transition involving one hydrogen moving more than the other. We finally associate orientational decorrelations to the transitions between stable states, excluding $C_1$ transitions that do not change the orientation of the molecule. In Extended Data Fig. 7d we report the transition probabilities between stable states in one or two

steps. We ignore higher order transitions that involve more than one intermediate state (such as, for example, S → E → E → S) as their probability is negligible (less than $10^{-5}$). We observe that the most frequent transition between stable states corresponds to the $C_4$ rotations, which are around twice the contribution deriving from $C_3$ rotations and more than one order of magnitude compared to $C_2$ rotations. Thus, these results justify the adoption of a $C_4$ model to describe the rotational relaxation of water-plastic phase observed in the experiments.

## Data availability

Raw neutron data were generated at the ILL (Grenoble, France) large-scale facility and are publicly available from the ILL depository at https://doi.org/10.5291/ILL-DATA.LTP-6-6 (ref. 44), https://doi.org/10.5291/ILL-DATA.DIR-195 (ref. 45) and https://doi.org/10.5291/ILL-DATA.5-24-620 (ref. 54). Molecular dynamics simulation data are available at Figshare (https://doi.org/10.6084/m9.figshare.c.7441660.v1)[57]. All other data used in this study are available from the corresponding author upon request.

## Code availability

The codes used within the molecular dynamics simulations are available at Figshare (https://doi.org/10.6084/m9.figshare.c.7429996.v1)[66].

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

**Acknowledgements** We acknowledge the Institut Laue-Langevin (ILL) for providing beamtime on IN6-SHARP and IN5. We acknowledge assistance from C. Payre and J. Maurice with the high-pressure setup and from J. Halbwachs on IN5. We thank A. Polidori and F. Alabarse for participating in the experiments and preliminary analysis. We also thank T. C. Hansen for help with the measurements on D20 and Benoit Baptiste for help with the XRD experiments. J.R. acknowledges support from the European Research Council grant no. DLV-759187. L.E.B. acknowledges the financial support by the European Union—NextGenerationEU (grant no. PRIN N. F2022NRBLPT), grant no. ANR-23-CE30-0034 EXOTIC-ICE and the Swiss National Fund (FNS) under grant no. 212889. J.R. and F.S. acknowledge support by ICSC – Centro Nazionale di Ricerca in High Performance Computing, Big Data and Quantum Computing, funded by European Union – NextGenerationEU.

**Author contributions** L.E.B. designed the research. L.E.B., M.R., A.T., F.M. and U.R. wrote the manuscript. M.R. prepared the figures, with inputs from L.E.B. The QENS experiments were performed by L.E.B., U.R., R.G., S.K., M.M.K. and J.O. M.R. and L.A. performed the XRD experiments. U.R., R.G. and S.K. performed the neutron diffraction experiments. M.R. performed the QENS data analysis with inputs from L.E.B., U.R., M.M.K and J.T. L.A. performed the XRD data analysis with inputs from S.K. A.T. performed the molecular dynamics simulations and data analysis with guidance from F.M. J.R. and F.S. All authors discussed the results and the manuscript.

**Funding** Open access funding provided by EPFL Lausanne.

**Competing interests** The authors declare no competing interests.

**Additional information**
**Correspondence and requests for materials** should be addressed to Livia Eleonora Bove.

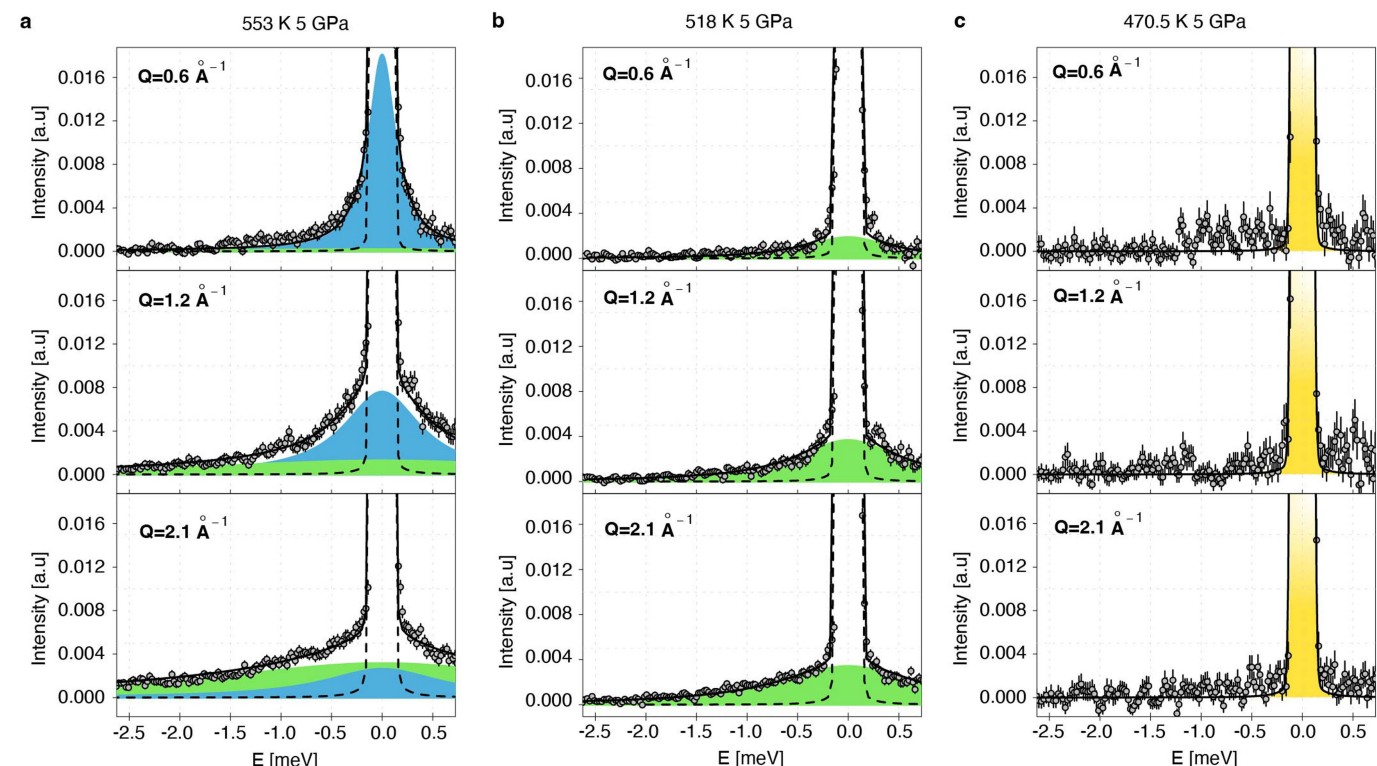

**Extended Data Fig. 1 | Experimental data along the 5.5 GPa isobar.** Fits to the QENS data in the liquid (553 K) (**a**), in plastic ice VII (518 K) (**b**) and in ice VII (470.5 K) (**c**). Error bars were calculated by the square root of absolute neutron count combined with the law of propagation of errors. The elastic contribution is shown as yellow area, the translational contribution, only visible in the liquid phase, is shown as blue area, the rotational contribution is shown as green area.

The instrumental resolution is shown as black dashed line while the total fit is represented as solid black line. A flat background related to instrumental background as well as the tails of the sample's and sample environment's vibration dynamics has been subtracted from the spectra. Fit results for the plastic and liquid data sets are shown in Extended Data Fig. 2.

**a**

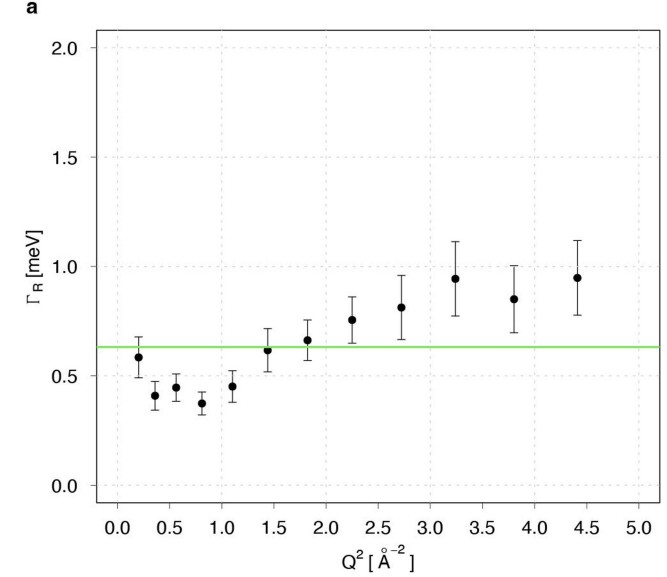

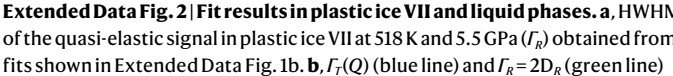

**b**

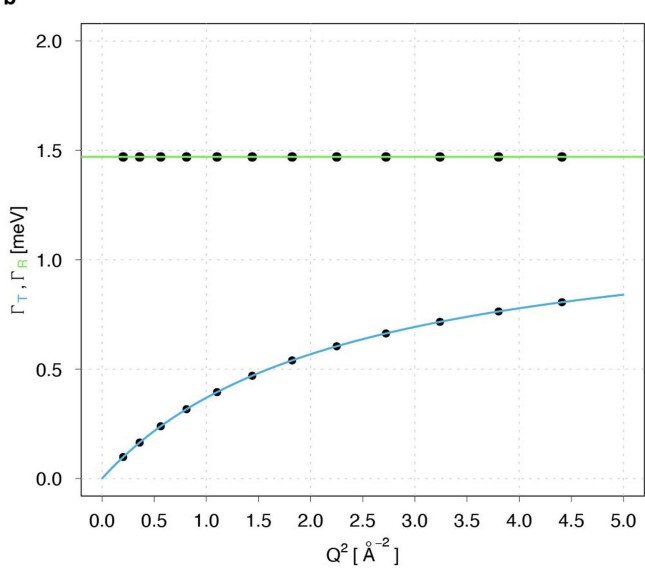

**Extended Data Fig. 2 | Fit results in plastic ice VII and liquid phases. a**, HWHM of the quasi-elastic signal in plastic ice VII at 518 K and 5.5 GPa ($\Gamma_R$) obtained from fits shown in Extended Data Fig. 1b. **b**, $\Gamma_T(Q)$ (blue line) and $\Gamma_R = 2D_R$ (green line) of equation (13), in the liquid phase, obtained from fits shown in Extended Data Fig. 1a. From these data we estimate the translational and rotational diffusion coefficients as: $D_T = 8.0 \pm 0.8\,10^{-5}\,cm^2/s$ and $D_R = 1.1 \pm 0.3\,ps^{-1}$ at 553 K and 5.5 GPa.

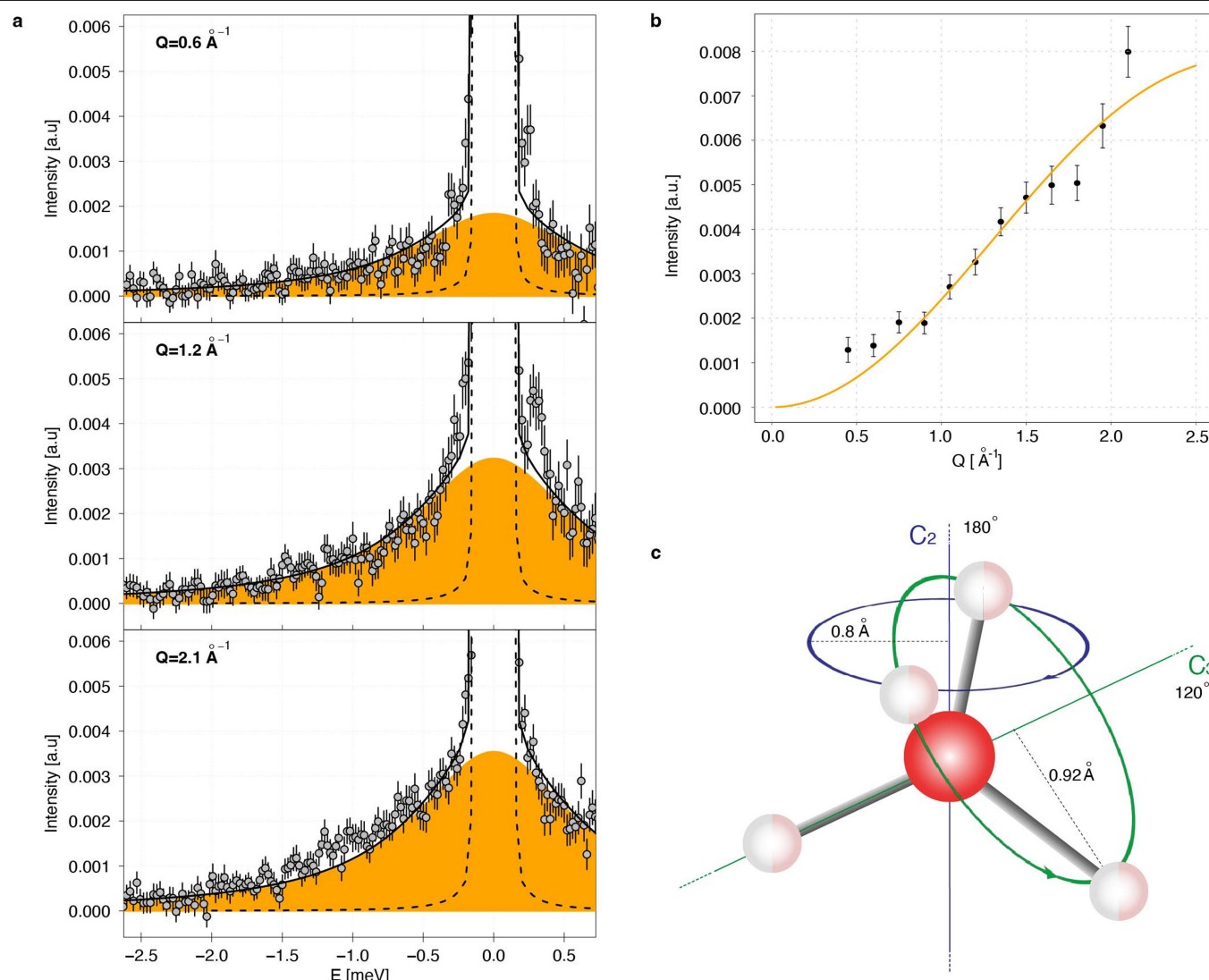

**Extended Data Fig. 3 | Reorientations around the $C_2$ or $C_3$ symmetry axes. a**, Examples of fits to the data at 518 K and 5.5 GPa, in plastic ice VII. Error bars were calculated by the square root of absolute neutron count combined with the law of propagation of errors. The orange area represents the fitted Lorentzian. **b**, Intensity of the rotational quasi-elastic Lorentzian as a function of $Q$, obtained as described in the *Fitting Procedure* section of *Methods*. Error bars represent standard deviations of the fitted parameters. **c**, Geometrical representation of the $C_2$ and $C_3$ symmetry axes.

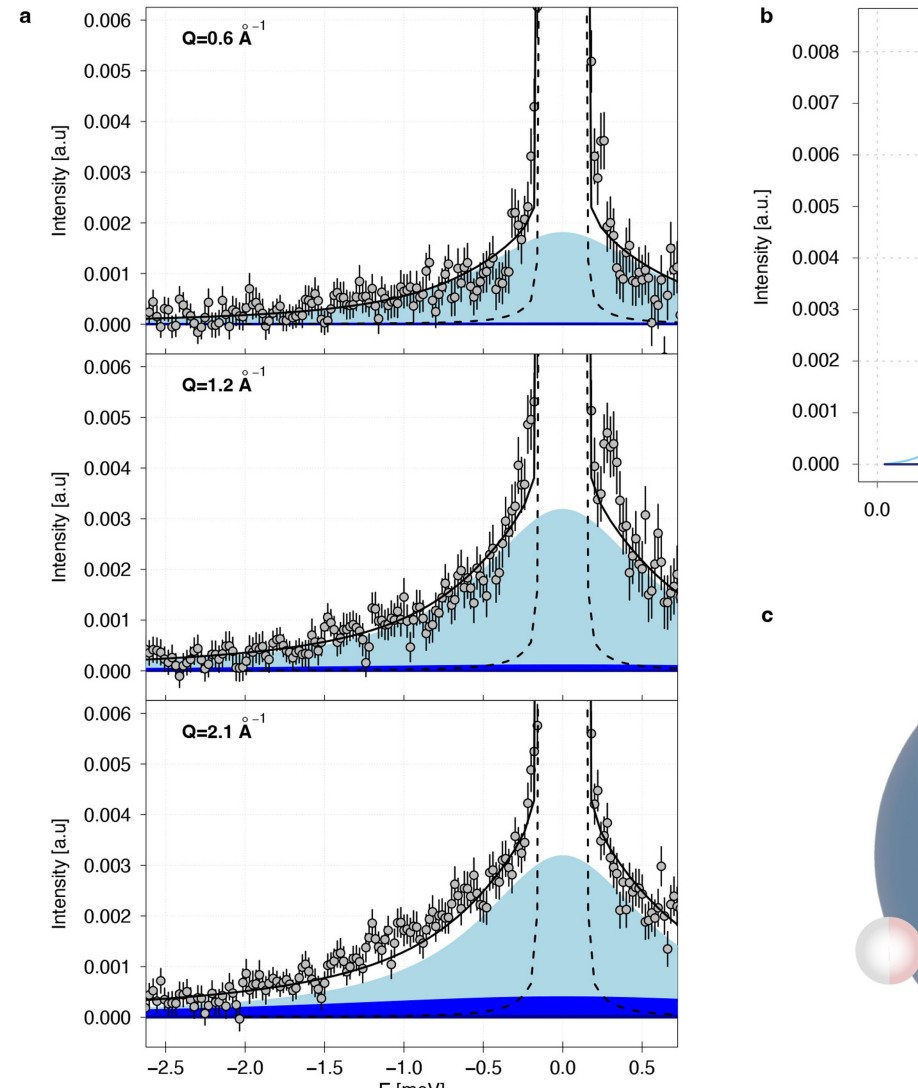

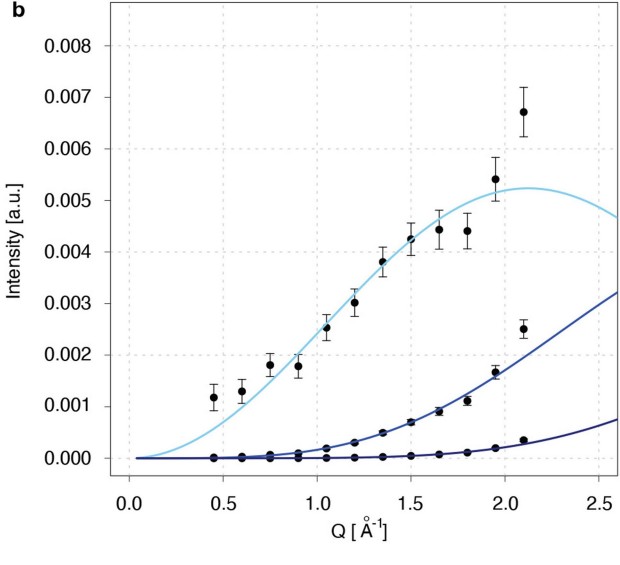

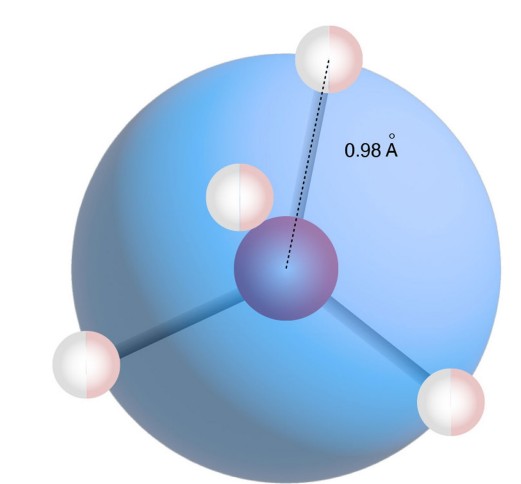

**Extended Data Fig. 4 | Isotropic rotations. a**, Examples of fits to the data at 518 K and 5.5 GPa, in plastic ice VII. Error bars were calculated by the square root of absolute neutron count combined with the law of propagation of errors. Areas represent the fitted Lorentzians. The relative intensities and widths of the three contributions are constrained to the model, intensity factor at each $Q$ value is left as free parameter. **b**, Intensity of the rotational quasi-elastic

Lorentzians as a function of $Q$, obtained as described in the *Fitting Procedure* section of *Methods*. Error bars represent standard deviations of the fitted parameters. For both **a** and **b**, light blue refers to the first term of the expantion in (6) ($l = 1$), blue to the second term ($l = 2$) and dark blue to the third term ($l = 3$) of the expansion. **c**, Geometrical representation of surface of the sphere.

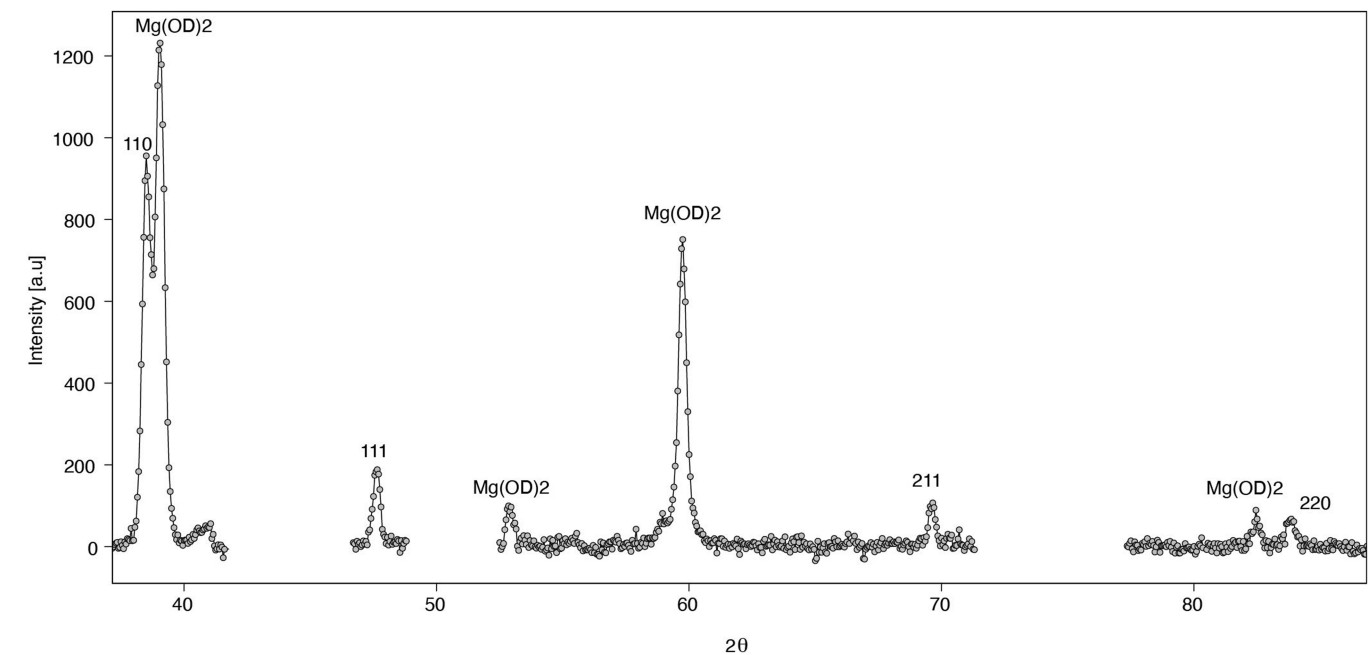

**Extended Data Fig. 5 | Neutron diffraction data at 500 K and 4.7 GPa.** MgO was used during the experiment in order to avoid single crystal formation, its reaction with deuterated water produced Mg(OD)$_2$ (for which the space group is $P\bar{3}m1$ and the unit cell volume is of about 38 Å$^3$ at this pressure). Visible reflections from ice VII (space group $Pn\bar{3}m$ with $a \approx 3.3$ Å) are indexed. The (111) reflection is visible in the thermodynamic region where we observe an active rotational dynamics. Excluded regions are those where we observe intense peaks from the Inconel gasket.

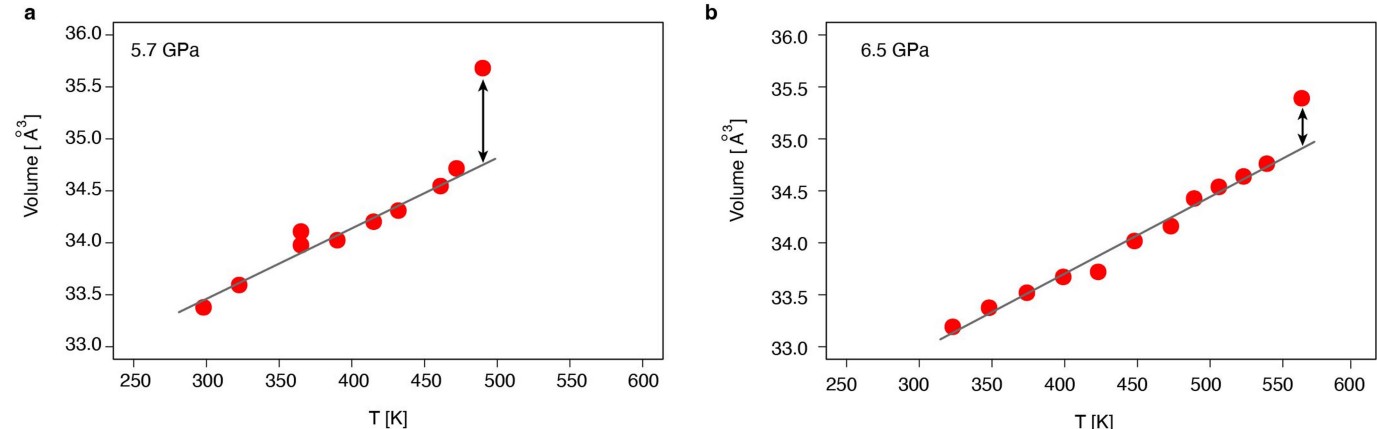

**Extended Data Fig. 6 | X-ray diffraction experiments.** Unit cell volumes as a function of temperature along two isobars at 5.7 GPa (**a**) and 6.5 GPa (**b**). Error bars are smaller than dots.

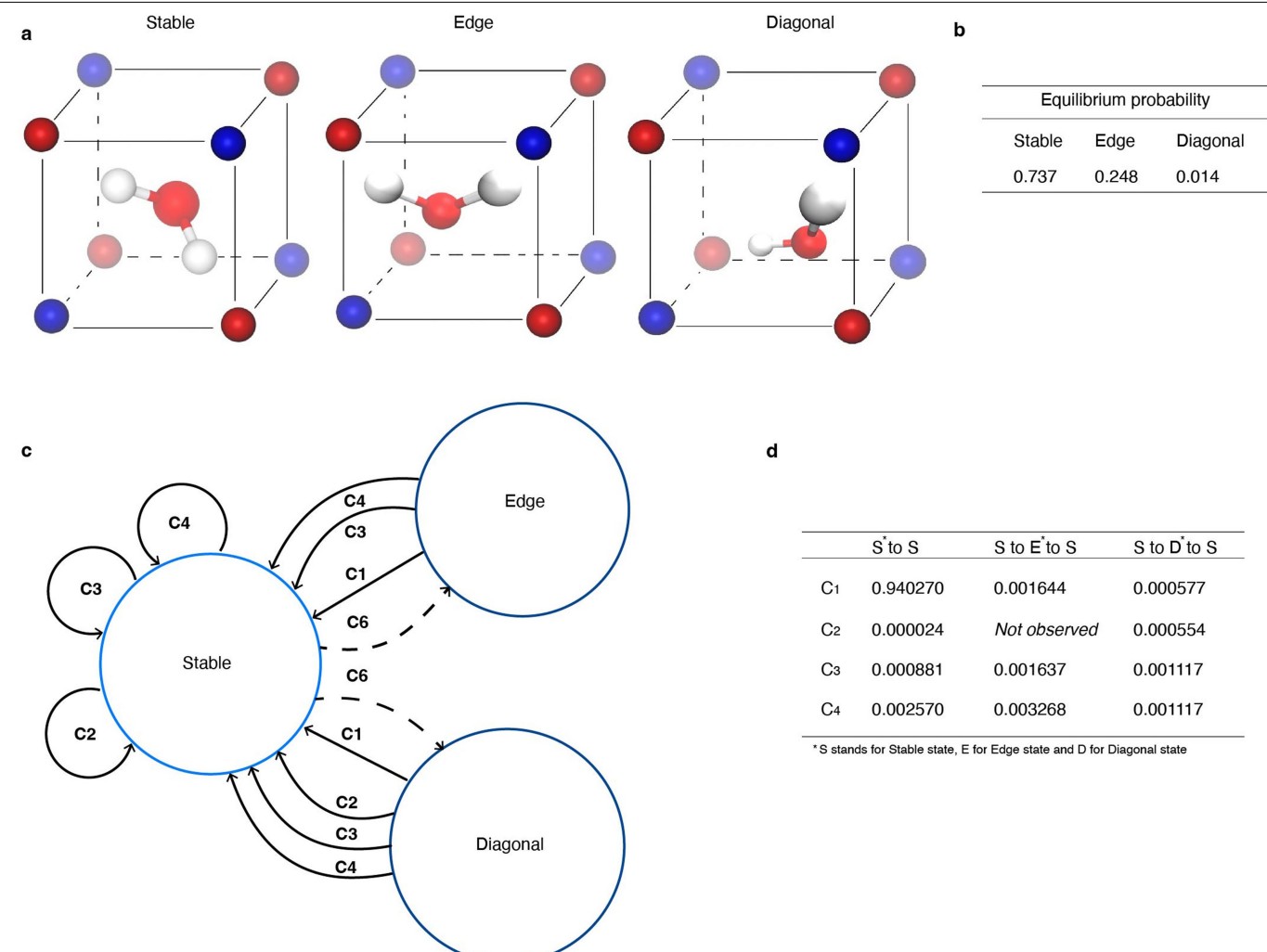

| Equilibrium probability | | |
|---|---|---|
| Stable | Edge | Diagonal |
| 0.737 | 0.248 | 0.014 |

|  | S* to S | S to E* to S | S to D* to S |
|---|---|---|---|
| $C_1$ | 0.940270 | 0.001644 | 0.000577 |
| $C_2$ | 0.000024 | *Not observed* | 0.000554 |
| $C_3$ | 0.000881 | 0.001637 | 0.001117 |
| $C_4$ | 0.002570 | 0.003268 | 0.001117 |

* S stands for Stable state, E for Edge state and D for Diagonal state

**Extended Data Fig. 7 | Markov Chain analysis. a**, Schematic representation of the possible configurations. The set of stable configurations presents the hydrogens oriented along the lattice's face diagonals, as represented in the figure on the left. The other two groups have, in one case, hydrogens looking at the edge of the lattice (Edge, central figure) or, in the latter case, orienting along the diagonals of the cubic lattice (Diagonal, right figure). **b**, Table with the probability at the equilibrium for each set of configurations. **c**, Schematic representation of main transitions among states, associated with their rotational description. **d**, Table with the probability of observing a specific kind of rotation that connects two Stable configurations (S).