## [Peer Review File · Nature]

Observation of Plastic Ice VII by Quasi-Elastic Neutron Scattering

Corresponding Author: Professor Livia Bove

Version 0:

Reviewer comments:

Referee #1

(Remarks to the Author)

Review of "Observation of Plastic Ice VII: Insights from Quasi Elastic Neutron Scattering" by Rescigno et al.

This paper describes a quasi-elastic neutron scattering experiment that observes plastic ice VII for the first time. Plastic ice was first predicted in 1999 by ab initio computer simulation and a number simulations since then have demonstrated the same phenomenon. But it has taken until now to observe this material experimentally, the reason being the need to perform the measurement at high temperature (~500K) and high pressure (~10GPa), so requiring specialist equipment and knowledge. Neutrons are used because of their ability to "see" inside substantial pressure cells, and also because the dynamics of the water molecule's hydrogen atoms cover a distinctly different time-scale and energy transfer window compared to all the materials that make up the pressure cell's body. As a result, both the onset of a plastic phase, where the water molecules can rotate but not translate, and the onset of melting, where they can both rotate and translate, gives rise to additional and readily identifiable quasi-elastic neutron scattering signals, which are not present when the water molecule merely librates in its lattice position. (See Figure 2 of the present paper.) Initial X-ray diffraction data (Supplementary Information) shows a sharp increase in the molar volume through the transition to the plastic phase.

The observation of plastic ice is important because of the likely implications, among other things, for ice planets in the Solar System and beyond. Plus, to my mind, observation of any new phase of ice or water deserves wide dissemination in its own right because of the fundamental importance of water in the the environment and in the Universe. As far as I can ascertain the work has been executed with considerable diligence and care, and, for all these reasons must be a strong contender for a publication in Nature.

The data are analysed in terms of an isotropic rotor model, and other models involving jump rotations around different axes. None of these is sufficiently distinct in the data to give an unequivocal indication of what type of motion is most likely, so the authors resort to a Molecular Dynamics study, which reveals that all of these jump motions can occur, but that those involving 90o jumps around the fourfold lattice axis rotations are the most common.

In my opinion the work has been described with considerable attention to detail and there is not a lot I can comment on. It should be readily accessible to someone with a modicum of knowledge about neutron scattering, which does not detract from its applicability to a wider audience. I do however have a couple of thoughts about some aspects of the paper that might help to strengthen its argument.

It struck me that the X-ray diffraction result mentioned above showing the sharp increase in molar volume at the plastic transition was sufficiently important to the overall argument that it should be brought into the main text: this transition is not simply some artefact of the quasi-elastic experiment, which might be a criticism that could be levelled at the paper from some quarters. The transition has direct consequences for other properties of the material, and the SI Figure 6 shows this very nicely. Unfortunately the data there are reported to be preliminary, but if that could be rectified some way prior to publication, I think it would strengthen the argument considerably.

I found Figure 1 to be somewhat chaotic. A diagram of the actual quasi-elastic experiment would have been helpful to non-specialist readers, since it is not a widely reported technique at this level. The references to the different sections in the caption and text should be in the order in which they are presented, a,b,c, and so on. I did wonder if the graph pairs, (a,b),

(d,e), and (f,g) could be presented in a different way (perhaps a separate figure?), namely a column format with b, e and g (the experimental data) in the first column, and a, d and f (simulation picture) in the second column, alongside the respective experimental graphic? The graphic h is not especially needed here as the equivalent is shown in Figure 2, but it could be part of a separate (first) graphic which describes the experiment and the state conditions under which the experiment was conducted. However the authors decide to present this information, I feel that presentation needs to be significantly improved over the present graphic.

I would add that many of the references lack the year of publication. I know this would have to be rectified prior to publication, but it is nonetheless quite annoying for the reviewer to not be able to see the year of publication easily!

Figures 3a, 6a and 7a lack a description of the shaded areas and their colours in the corresponding figure captions.

Figure 9 of the SI refers to equation 7, but I think this is meant to be equation 6?

In summary, I felt this to be a very worthwhile paper that would be of interest to a wide audience. The evidence for the plastic transition seems pretty solid, and this particular aspect of water and ice properties does not carry all the baggage and controversy that has accompanied studies of the meta-stable states of water. I think the editors should give it serious consideration for publication in Nature.

Referee #2

(Remarks to the Author)

Even though the existence of a plastic phase in ice VII is recognized by theoretical calculations, this report is the first experimental confirmation of the prediction. This is an important contribution to understanding the phase diagram of ice, particularly at the liquid-solid boundary.

Neutron quasi-elastic scattering is a well-established technique for characterizing molecular dynamics. The quality of the neutron measurements reported in this study were high. The dynamic models used to fit the experimental structure factors are appropriate. The statistical errors are presented. It is impressive that several (P, T) points were examined along the liquid-ice VII boundary, as illustrated in Fig. 1c.

Due to proton disorder, the oxygen atom of a water molecule is situated in an average tetrahedral environment. However, at each instance, there are two strong covalent and two weak hydrogen bonds. Thus, the water molecule in ice VII is not a free rotor. The conclusion of a hopping mechanism is not unexpected. Markov chain analysis of the trajectories computed from molecular dynamics (MD) calculations assuming rigid water molecules support the hopping mechanism.

The manuscript is well-written. Overall, the facts and discussions are clearly presented.

I have the following suggestions:

1. For completeness, it is desirable to compare the computed dynamic structure factor $S(q, \omega)$ with the corresponding experimental results, perhaps in the supplementary materials.
2. It was mentioned on page 13 that the fitted Debye-Waller factor of 0.78 \AA^2 at 3.5 GPa is in good agreement with the expected result. Please state the theoretical value. How about the agreement for other pressures and temperatures?
3. Can diffuse scattering due to the "hopping" deuterium be observed in the diffraction pattern?

Referee #1 (Remarks to the Author):

Review of “Observation of Plastic Ice VII: Insights from Quasi Elastic Neutron Scattering” by Rescigno et al.

This paper describes a quasi-elastic neutron scattering experiment that observes plastic ice VII for the first time. Plastic ice was first predicted in 1999 by ab initio computer simulation and a number simulations since then have demonstrated the same phenomenon. But it has taken until now to observe this material experimentally, the reason being the need to perform the measurement at high temperature (~500K) and high pressure (~10GPa), so requiring specialist equipment and knowledge. Neutrons are used because of their ability to “see” inside substantial pressure cells, and also because the dynamics of the water molecule’s hydrogen atoms cover a distinctly different time-scale and energy transfer window compared to all the materials that make up the pressure cell’s body. As a result, both the onset of a plastic phase, where the water molecules can rotate but not translate, and the onset of melting, where they can both rotate and translate, gives rise to additional and readily identifiable quasi-elastic neutron scattering signals, which are not present when the water molecule merely librates in its lattice position. (See Figure 2 of the present paper.) Initial X-ray diffraction data (Supplementary Information) shows a sharp increase in the molar volume through the transition to the plastic phase.

The observation of plastic ice is important because of the likely implications, among other things, for ice planets in the Solar System and beyond. Plus, to my mind, observation of any new phase of ice or water deserves wide dissemination in its own right because of the fundamental importance of water in the the environment and in the Universe. As far as I can ascertain the work has been executed with considerable diligence and care, and, for all these reasons must be a strong contender for a publication in Nature.

The data are analysed in terms of an isotropic rotor model, and other models involving jump rotations around different axes. None of these is sufficiently distinct in the data to give an unequivocal indication of what type of motion is most likely, so the authors resort to a Molecular Dynamics study, which reveals that all of these jump motions can occur, but that those involving 90o jumps around the fourfold lattice axis rotations are the most common.

In my opinion the work has been described with considerable attention to detail and there is not a lot I can comment on. It should be readily accessible to someone with a modicum of knowledge about neutron scattering, which does not detract from its applicability to a wider audience. I do however have a couple of thoughts about some aspects of the paper that might help to strengthen its argument.

We thank the referee for carefully evaluating our work, for appreciating the value of the results presented and for the pertinent comments.

It struck me that the X-ray diffraction result mentioned above showing the sharp increase in molar volume at the plastic transition was sufficiently important to the overall argument that it should be brought into the main text: this transition is not simply some artefact of the quasi-elastic experiment, which might be a criticism that could be levelled at the paper from some quarters. The transition has direct consequences for other properties of the material, and the SI Figure 6 shows this very nicely. Unfortunately the data there are reported to be preliminary, but if that could be rectified some way prior to publication, I think it would strengthen the argument considerably.

We agree with the referee that the observation of a 2-3% volume change, if confirmed, is highly significant and warrants further attention. However, as now more clearly explained in the text, these measurements were conducted on a laboratory Rigaku diffractometer. The primary issue is that, as we approached the melting curve, the sample formed oriented monocrystals—this behavior is well known and widely reported in the literature. Consequently, we were unable to record a reliable diffraction signal, despite visually observing the presence of crystals in the sample.

Moreover, to obtain sufficiently precise measurements to estimate both the pressure from the gold calibrant and the lattice volume from ice VII, we needed to integrate over a minimum of one hour. At high temperatures, the pressure stability was suboptimal due to DAC relaxation, requiring us to repeat the measurements multiple times to acquire stable data. This approach is not ideal for estimating potential hysteresis effects.

Following the referee's suggestion, we conducted a new trial with the same experimental setup but using different pressure calibrants. However, this did not result in any significant improvement to the data. We believe that more systematic measurements using a synchrotron radiation source are necessary. A more focused and intense beam would likely yield signals from single crystallites in a much shorter time, thereby ensuring stable pressure (P) conditions. To this end, we have proposed an experiment for the current proposal cycle, and if successful, the findings will be detailed in a subsequent report.

We can confidently exclude QENS artifacts, as the experiments were repeated across three beamtimes on two different spectrometers with varied sample chamber geometries, all yielding consistent results. Additionally, the observed transitions—from crystal to plastic to liquid along isobars, and from liquid to plastic to crystal along isotherms—clearly demonstrated the unblocking of rotational dynamics and the freezing of translational dynamics within the same sample. This minimized potential changes in background noise or scattering effects between the three states.

Finally, the quantitative analysis of the QISF intensity is particularly noteworthy. It points to the correct molecular radius of the water molecule and the expected tetrahedral geometry, a stringent test that is rarely conducted in QENS experiments. This further strengthens our confidence in the results.

We opted for moving the XRD results from the supplementary information to the methods section and citing such results in the main text.

I found Figure 1 to be somewhat chaotic. A diagram of the actual quasi-elastic experiment would have been helpful to non-specialist readers, since it is not a widely reported technique at this level. The references to the different sections in the caption and text should be in the order in which they are presented, a,b,c, and so on. I did wonder if the graph pairs, (a,b), (d,e), and (f,g) could be presented in a different way (perhaps a separate figure?), namely a column format with b, e and g (the experimental data) in the first column, and a, d and f (simulation picture) in the second column, alongside the respective experimental graphic? The graphic h is not especially needed here as the equivalent is shown in Figure 2, but it could be part of a separate (first) graphic which describes the experiment and the state conditions under which the experiment was conducted. However the authors decide to present this information, I feel that presentation needs to be significantly improved over the present graphic.

We have revised Figures 1 and 2 in the main text in response to the referee's suggestions. Specifically, we simplified Figure 1, which now consists of three panels: the phase diagram showing the measured thermodynamic points, a schematic of the quasi-elastic neutron scattering experiment, and an example of the dynamical structure factor (formerly panel h of Figure 1). In Figure 2, we now display the experimental data alongside MD simulation snapshots, organized in separate columns as recommended by the referee.

I would add that many of the references lack the year of publication. I know this would have to be rectified prior to publication, but it is nonetheless quite annoying for the reviewer to not be able to see the year of publication easily!

We apologize for the inconvenience. During the review process, we reduced the number of references and added the missing publication years to all references.

Figures 3a, 6a and 7a lack a description of the shaded areas and their colours in the corresponding figure captions.

We have modified the captions of figures 3, 6 and 7 and stated explicitly what the areas and colors refer to.

Figure 9 of the SI refers to equation 7, but I think this is meant to be equation 6?

We thank the referee for pointing out this inconsistency. We have modified the figure legend, the reference to the correct equation is now in the caption of the figure.

In summary, I felt this to be a very worthwhile paper that would be of interest to a wide audience. The evidence for the plastic transition seems pretty solid, and this particular aspect of water and ice properties does not carry all the baggage and controversy that has accompanied studies of the meta-stable states of water. I think the editors should give it serious consideration for publication in Nature.

We thank the referee for the positive feedback and for the constructive comments and suggestions.

Referee #2 (Remarks to the Author):

Even though the existence of a plastic phase in ice VII is recognized by theoretical calculations, this report is the first experimental confirmation of the prediction. This is an important contribution to understanding the phase diagram of ice, particularly at the liquid-solid boundary.

Neutron quasi-elastic scattering is a well-established technique for characterizing molecular dynamics. The quality of the neutron measurements reported in this study were high. The dynamic models used to fit the experimental structure factors are appropriate. The statistical errors are presented. It is impressive that several (P, T) points were examined along the liquid-ice VII boundary, as illustrated in Fig. 1c.

Due to proton disorder, the oxygen atom of a water molecule is situated in an average tetrahedral environment. However, at each instance, there are two strong covalent and two weak hydrogen bonds. Thus, the water molecule in ice VII is not a free rotor. The conclusion of a hopping mechanism is not unexpected. Markov chain analysis of the trajectories computed from molecular dynamics (MD) calculations assuming rigid water molecules support the hopping mechanism.

The manuscript is well-written. Overall, the facts and discussions are clearly presented.

We thank the reviewer for the positive comments on our manuscript.

I have the following suggestions:

1. For completeness, it is desirable to compare the computed dynamic structure factor $S(q, \omega)$ with the corresponding experimental results, perhaps in the supplementary materials.

We thank the reviewer for the valuable suggestion. In response, we have included a comparison between the experimental and simulated dynamical structure factors at different Q values in the supplementary materials. Overall, we observe good agreement, particularly at higher Q values. A dedicated section discussing this comparison has been added to the supplementary materials.

2. It was mentioned on page 13 that the fitted Debye-Waller factor of 0.78 \AA^2 at 3.5 GPa is in good agreement with the expected result. Please state the theoretical value. How about the agreement for other pressures and temperatures?

To the best of our knowledge, Debye-Waller data for Ice VII at higher temperatures are not available in the literature, which is why the temperature and pressure dependence

was not addressed in the submitted manuscript. After reviewing our numerical predictions, we have added a column in Table 1 showing the mean square displacements under different thermodynamic conditions, which are in good agreement with the experimental results.

3. Can diffuse scattering due to the “hopping” deuterium be observed in the diffraction pattern?

We do not observe diffuse scattering due to the hopping deuterium or hydrogen in the diffraction patterns.

In general, disorder can be observed (and modeled) in diffraction data, but this typically requires high-quality single-crystal diffraction measurements. However, due to the constraints of high-pressure environments—such as limited angular opening and small sample volumes—we believe this approach is at the limit of the capabilities of state-of-the-art Neutron Diffraction techniques. However an attempt can be made in a future proposal. That said, it may be also possible to observe diffuse scattering using X-ray diffraction at synchrotron facilities. As mentioned in our response to Referee 1, we have submitted a proposal for the next round at ESRF to undertake this challenging experiment.